# Identification of Biomarker Volatile Organic Compounds Released by Three Stored-Grain Insect Pests in Wheat

**DOI:** 10.3390/molecules27061963

**Published:** 2022-03-17

**Authors:** Lijun Cai, Sarina Macfadyen, Baozhen Hua, Haochuan Zhang, Wei Xu, Yonglin Ren

**Affiliations:** 1State Key Laboratory of Ecological Pest Control for Fujian/Taiwan Crops, Institute of Applied Ecology, Fujian Agriculture and Forestry University, Fuzhou 350002, China; cai-lijun@live.cn; 2Key Laboratory of Plant Protection Resources and Pest Management, Ministry of Education, Northwest A&F University, Xianyang 712100, China; huabz@nwafu.edu.cn; 3Agriculture and Food, Commonwealth Scientific and Industrial Research Organisation, Acton, ACT 2601, Australia; sarina.macfadyen@csiro.au; 4College of Science, Health, Engineering and Education, Murdoch University, Murdoch, WA 6150, Australia; 32985118@student.murdoch.edu.au

**Keywords:** volatile organic compounds (VOCs), headspace solid-phase microextraction (HS-SPME), SPME-GC-MS, stored-product insect pests, *Tribolium castaneum*, *Rhyzopertha dominica*, *Sitophilus granarius*

## Abstract

Monitoring and early detection of stored-grain insect infestation is essential to implement timely and effective pest management decisions to protect stored grains. We report a reliable analytical procedure based on headspace solid-phase microextraction coupled with gas chromatography–mass spectrometry (HS-SPME-GC-MS) to assess stored-grain infestation through the detection of volatile compounds emitted by insects. Four different fibre coatings were assessed; 85 µm CAR/PDMS had optimal efficiency in the extraction of analytes from wheat. The headspace profiles of volatile compounds produced by *Tribolium castaneum* (Herbst), *Rhyzopertha dominica* (Fabricius), and *Sitophilus granarius* (Linnaeus), either alone or with wheat, were compared with those of non-infested wheat grains. Qualitative analysis of chromatograms showed the presence of different volatile compound profiles in wheat with pest infestation compared with the wheat controls. Wheat-specific and insect-specific volatile compounds were identified, including the aggregation pheromones, dominicalure-1 and dominicalure-2, from *R. dominica*, and benzoquinones homologs from *T. castaneum*. For the first time, the presence of 3-hydroxy-2-butanone was reported from *S. granarius*, which might function as an alarm pheromone. These identified candidate biomarker compounds can be utilized in insect surveillance and monitoring in stored grain to safeguard our grain products in future.

## 1. Introduction

Maintenance of quality parameters in stored grain over extended periods is of critical importance to global food security [1,2]. Annually, around one-third of food production, about 1.3 billion tonnes worth, approximately, US $1 trillion, is lost after harvest operations to infestation by pests and microorganisms [3]. Stored grains can be damaged by multiple insect pest species including *Tribolium castaneum* (Herbst), *Rhyzopertha dominica* (Fabricius), and *Sitophilus granarius* (Linnaeus), three of the most damaging stored-grain insect pests around the world. *R. dominica* and *S. granarius* are notorious primary pests of stored products that live in whole grain kernels, while *T. castaneum* is commonly considered as a secondary pest, appearing in stored products after the primary infestation. These species not only cause direct damage to stored grains through feeding, but also deteriorate grain quality [4]. Reliable and simple methods of early detection of these pests in stored grains are critical to their control throughout the supply chain, with implications for storage and the multi-billion-dollar international trade in grains.

To detect stored-product insects in grains, current techniques include near-infrared spectroscopy [5], digital imaging [6,7,8,9], and aroma sensing (techniques such as electronic nose, or “eNose”) [9,10,11,12,13]. However, these methods are either labour intensive, susceptible to environmental influences, inefficient in detecting immature stages of insects and internal infestation, or not sensitive enough [14]. Novel methods of sample collection, insect detection, and data analysis should be developed to report the real-time quality of stored grains, allowing fast and appropriate management decisions. A potential detection approach is to investigate the air samples from a grain mass to detect specific volatile compounds (VOCs) released by damaging insects, which can be biomarkers for monitoring grain infestation.

Since its original use for the isolation and identification of volatile chemicals in the food industry, headspace solid-phase micro-extraction (HS-SPME), coupled with a gas chromatograph–flame ionisation detection (GC-FID) or gas chromatograph–mass spectrometry (GC-MS) technique, has been widely used to examine pheromones and other volatile secretions of coleopteran insects [15,16,17,18,19]. For example, this technique has been utilized to identify the aggregation pheromones, dominicalure 1 and dominicalure 2, of *R. dominica*, and how pest infestation influences the volatile profiles of grains, as well as volatile compounds of wheat alone [4]. The compound 4,8-dimethyldecanal (DMD), which is a well-known male aggregation pheromone in Tenebrionidae, and other volatile metabolites, such as benzoquinones in *Tribolium* spp., are potential indicators of infestation by these insects [20,21]. Conversely, little attention has been paid to investigating other volatile chemicals of *S. granarius*. A well-known exception is the male-produced aggregation pheromone identified as 1-ethylpyropyl (2R, 3S)-2-methyl-3-hydroxypentanoate [22,23], commonly known as 2R, 3S-sitophilate. In addition, the HS-SPME technique has also been reported in detecting musty–earthy off-odours in wheat [24].

The aim of this study was to identify candidate VOCs released by pest infestation in stored grains using the HS-SPME-GC-MS technique. The volatile compound profiles obtained from non-infested grains were compared with those of grains infested with *R. dominica*, *T. castaneum*, or *S. granarius*. The volatile profiles of the three pest species alone were also examined. This study verifies changes in the relative responses of the target compounds among treatments and identified specific volatiles that can be used as potential indicators of pest infestations in stored grains. 

## 2. Results

### 2.1. SPME Fibre Selection

Air only, as a blank check, was tested for each fibre as well, but they did not demonstrate any meaningful peaks, or the peaks were too tiny when compared with the signals released by the treatments such as wheat and insects. The chromatograms obtained from the four fibres tested are shown in Figure 1. The CAR/PDMS fibre showed the best extraction efficiency of the analytes from wheat because it detected the highest number of peaks and most peaks exhibited sharply. For example, we compared the identified compound peaks that were higher than 250 mV and were detected by each fibre. CAR/PMDS detected a total of 14 peaks while the other fibres detected less than two peaks. When the retention time (RT) was <5 min, CAR/PMDS detected more and larger peaks than the other three fibres. This result accords with the guide for fibre selection provided by the manufacturer for relatively small molecular weight compounds. Therefore, the 85 μm CAR/PDMS fibre was selected for subsequent method optimization experiments.

### 2.2. Selection of Sampling Time

The general volatile patterns of headspace tested at different time points are quite similar (Figure 2). The main peaks identified for wheat only were acetone, methanol, ethanol, 2,3-butanedione, 2-butanol, hexanal, 2-methyl-1-propanol, 1-butanol, 1-penten-3-ol, 3-methyl-1-butanol, 1-pentanol, 3-methyl-2-buten-1-ol, 1-hexanol, and 1-pentadecene, numbered from 1 to 14, respectively (Figure 2a). Other common compounds, such as dimethyl sulphide and ethyl formate, were also detected and identified but in quite low amounts. The peak areas of the main compounds detected are very close among the three time points (at 2–4, 22–24, and 46–48 h) (Table 1), except for acetone (peak 1) (Figure 2b).

### 2.3. Single-Species Experiments

For all three insect species tested, treatments with insects only produced fewer volatile compounds when compared with the wheat-only controls. Each species had its own diagnostic volatile compounds which could be visually distinguished from each other (Figure 3a). *T. castaneum* adults only produced three significant peaks, namely T1, T2, and T3, the most abundant ions of which showed that they might be benzoquinones. The volatile pattern was the same with mixed gender samples or with adult male only or adult female only. 

*R. dominica* adults exhibited two typical detectable peaks, R6 and R7, which were identified as domimicalure 2 (D2) and dominicalure 1(D1), respectively (Figure 3b). Peak 14, identified as 1-pentadecane, might be associated with traces of culture medium. Peak intensities of D1 and D2 were very low and did not increase over time. Besides the characteristic compounds of *R. dominica*, acetone, ethanol and 1-pentadecene (Peaks 1, 3, and 14, Figure 2) were also detected but peak heights were very small. 

Detectable and reproducible signals from *S. granarius* adults could only be obtained after being held in a flask for over 40 h. The peaks characterizing *S. granarius* volatile emissions were marked as peaks S1, S2, and S3. Peak S1 was identified as 3-hydroxy-2-butanone (Figure 3c). In addition, several volatile compounds (peaks 1, 4, 12–14) originating from wheat were detected as trace amounts, and probably resulted from the wheat culture medium. These are numbered in Figure 2.

The volatile patterns of wheat only and wheat plus insects, for the three species tested, showed distinct patterns (Table 2). For wheat plus *T. castaneum*, the chromatogram turned out to be a simple combination of that of wheat only and that of *T. castaneum* only, but *T. castaneum* generated relevant high boiling point Benzene, 1-ethoxy-4-isothiocyanato, stearic acid, and one unknown compound (Table 2). Wheat plus *R. dominica* showed the most complex pattern; it not only had all the peaks from both wheat alone and *R. dominica* alone, but also seven new VOCs, such as ethyl acetate, 11-methylpentacosane, palmitic acid, 1-pentadecene, dominicalure 1, apparent homologs of dominicalure 1, and one unknown compound which could only be detected when *R. dominica* adults were added to wheat in the same flask (Peaks R1~R5 and Peaks R8~9 in Table 2). In addition, when *R. dominica* and wheat were combined, R6 (dominicalure 1) and R7 (dominicalure 2) increased dramatically and dwarfed any other components. In the case of wheat plus *S. granarius*, most of the typical VOCs of this species disappeared except VOC of 1-pentadecene, which was 2–3 folds enhanced in peak intensity than in wheat alone or in *S. granarius* alone (Table 2). 

## 3. Discussion

The HS-SPME technique coupled to GC-MS was utilized to investigate volatiles in infested stored grains for biomarkers of insect infestation. The CAR/PDMS coating fibre was recommended for the extraction of volatiles in grains infested with the tested species *R. dominica*, *T. castaneum*, and *S. granarius*. Previous research showed that the CAR/PDMS fibre had the highest efficiency of absorption of the overall volatile organic compounds from disturbed *T. castaneum* [21,25]. Our results confirmed that the CAR/PDMS fibre was efficient in extracting volatiles emitted by one single individual of *T. castaneum*. Whether using only one single individual or 20, the signals were detectable in the first 2 to 4 h after secretion (data not shown). In this study, only *R. dominica*, *T. castaneum*, and *S. granarius* adults were used to identify VOCs. Insect eggs, larvae, or pupae may produce different biomarker volatile compounds, which needs further investigation in future.

The samples containing wheat only showed the presence of the common volatile compounds, including dimethyl sulphide, acetone, methanol, ethanol, and butanol, which have been reported in previous research on clean wheat grains [26,27]. Nevertheless, fewer constituents were observed in our investigations, which were conducted under controlled laboratory temperatures, as compared with heated or distilled methods [26]. In addition, we found that more simple compounds in the form of alcohols were obtained in whole grains than in polished–graded wheat flours and distillers’ grains, in which aldehydes and ketones are more common [15,26]. The increase of acetone levels (shown as increased peak areas) over time might be due to the natural accumulation of respiration products. 

Volatile compounds of *T. castaneum* were abundant in peak intensity but were of a limited variety. Secretions of benzoquinones and the male-produced aggregation pheromone 4,8-dimethyldecanal (DMD), by *T. castaneum* have been well studied [20,21,28]. The extractable level and detected amounts vary between sampling methods and assay design. Our data showed that using the CAR/PDMS fibre was an efficient method for detecting benzoquinones. However, further identification is needed to confirm whether they are methyl-1,4-benzoquinone (MBQ) and ethyl-1,4-benzoquinone (EBQ), as reported in previous studies [20,21,29]. Similar to a previous study [21], no DMD was detected in our investigations. It was reported that amounts of DMD varied from 0.7 ng/male in a 4-day collection period with SPME analysis [20], to over 600 ng/day per male by using a Super Q column [30]. However, both two studies sexed and maintained adult *T. castaneum* separately for 30 days. Here, no DMD was detected in our experiment, which utilized unsexed *T. castaneum*. It was shown that high population densities have negative effects on DMD production [31], which were not apparent when food was absent. Nevertheless, DMD was not detected even when *T. castaneum* was combined in a sealed flask with wheat. The presence of primary pest species, such as *R. dominica* and *S. granarius*, in wheat might facilitate the release of aggregation pheromones in secondary pests. 

The typical peaks R6 and R7 that characterise the presence of *R. dominica* increased over time when they were added to wheat grains. This finding is consistent with previous research which concluded that *R. dominica* releases aggregation pheromones when they identified food resources. However, it has been reported that the pheromone released by this species was highest in late afternoon and the variance of actual quantities could reach up to 10 times more among individuals [32]. Here, we used a large population density of 80 adults per flask, so differences between individuals were not examined. In our work, we reported only a few compounds compared with around 90 compounds identified previously [4]. In the latter work, *R. dominica* was killed and removed before the headspace collection of samples. A long infestation period of 25 days, metabolites of the insects, and microbial contamination might also have contributed to the larger number of compounds detected. Moreover, heating can increase mass transfer, and this can lead to a greater recovery of higher molecular weight volatile compounds. Our experiments were carried out at room temperature and in a relatively natural storage environment. This is a useful way to avoid artificial reactions of volatile reagents in the headspace, but it is less efficient in completely absorbing all the volatiles, particularly those of low volatility. In another study, with both polar and non-polar columns at 25 °C, 114 compounds were detected in the headspace of three different samples, healthy wheat, *R. dominica*, and wheat, with *R. dominica*, indicating that the use of both polar and non-polar columns is essential to capturing the full range of VOCs produced [33]. 

*S. granarius* adults presented the most characteristic peaks compared with the other two pest species tested, besides some components in common with those of wheat. Here, for the first time, the compound 3-hydroxy-2-butanone was found in *S. granarius*. It has been reported as one of the three male-produced sex pheromones in the lobster cockroach, *Nauphoeta cinerea* (Olivier), and functions in agonistic interaction [34]. We suggest that this compound may be an alarming pheromone among *S. granarius* adults when they are under crowded and starving conditions. Further bioassay or behavioural function studies are needed to confirm its bio-function. To our knowledge, this is the first study examining the volatile compounds of grains infested by *S. granarius*. Moreover, 1-pentadecene increased two-fold in the headspace of wheat plus *S. granarius*, and it appeared to be a potentially useful indicator of infestation by *S. granarius*. It would be of interest to determine whether the compounds coming from *S. granarius* adults alone (S2 and S3) are absorbed by wheat or are not released once insects have found a food resource. In a recent study, wheat bran, with a low amount of 1-pentadecene added, was more attractive than wheat bran alone to *T. castaneum*, whereas higher concentrations of 1-pentadecene were repellent [35]. Sensitivity and reproducibility are the critical factors to consider in developing detection techniques for grain infestation. The fact that bulk grains are good sorbents to weak physical signals in the form of electrical, magnetic, nuclear, acoustic, or thermal energy is one of the greatest challenges for detection techniques. Our study showed that detectable signals could be recorded for only a single adult *T. castaneum*. However, further work needs to be conducted to determine the detection limits of this technique for *R. dominica* and *S. granarius*, mainly in non-laboratory conditions. 

In summary, a simple and reliable detection method for stored-product pests is urgently needed in the grain industry considering the great annual losses during storage due to pest infestations. The results, here, demonstrated that SPME-GC-MS is an efficient technique and has the potential to identify volatile chemical compounds of different insect species which can be used as early warning indicators of insect presence or infestation. A complete VOC profile of all the main stored-product pests would be necessary for future rapid diagnosis and monitoring to draw practical conclusions.

## 4. Materials and Methods

### 4.1. Insect Materials

*Tribolium castaneum* (TC4), *Rhyzopertha dominica* (RD2), and *Sitophilus granarius* (LG2) were obtained from the CSIRO Entomology Culture Collection, Canberra, Australia [36]. Single-species insect cultures were set up using adults and reared in 2 L glass culture jars under dark laboratory conditions at 30 °C and 60% relative air humidity. Two grams of *T. castaneum* were cultured on 734 g of whole organic wheat flour and 66 g of brewer’s yeast at a ratio of 12:1 (*w*/*w*). *R. dominica* (1.5 g) were cultured on 721 g of organic whole wheat grains with one cup of wheat flour on top. *S. granarius* (2.8 g) were cultured on 800 g of organic whole wheat grain. The adult insects were cultured for 4–5 weeks until adults of the next generation emerged. 

### 4.2. SPME Fibre Selection

Four SPME fibres with different coatings were tested and compared with the extraction efficiency of volatile compounds: 85 μm Carboxen/Polydimethylsiloxane (CAR/PDMS) (Sigma-Aldrich, Sydney, Australia, Cat. 57334-U), 65 μm Polydimethylsiloxane/Divinylbenzene (PDMS/DVB) (Sigma-Aldrich, Sydney, Australia, 57326-U), 85 μm polyacrylate (PA) (Sigma-Aldrich, Sydney, Australia, 57304), and 100 μm Polydimethylsiloxane (PDMS) (Sigma-Aldrich, Sydney, Australia, 57300-U). All fibres were conditioned prior to use in accordance with the manufacturer’s recommendations. The wheat variety ‘Rosella’ was used for testing the fibres. Grain of this variety was stored at 10 °C until used. Two hundred grams of wheat grain was added to a 250 mL Erlenmeyer flask (Bibby Sterilin, Staffordshire, UK, Cat. No. FE 250) and sealed with a Teflon-faced septum lid. There were three replicates for each treatment and the control. Sample collection was performed manually using a SPME fibre holder. The coated fibre was exposed to the upper portion of the headspace of wheat grains, just below the septum, for 3 h without agitation. The compounds were then desorbed from the fibre in the GC injector at 250 °C for 5 min under splitless mode.

### 4.3. Selection of Sampling Time

To standardize sampling time according to volatile emissions for the different insects used (treatments with insects) and monitor possible changes in volatile compound profiles in 250 mL sealed flasks over 48 h (treatment with wheat only), the fibres were suspended in the flask above the insects or wheat, or wheat plus insects, and out of reach of the insects to extract the volatiles for 3 h. According to our pre-experiments, the collection time used in this research was selected as the earliest timepoint that each species can give out a consistent and stable release (Table 1). For 20 g of homogenized wheat only, headspace samples were collected at 2–4, 22–24, and 46–48 h, respectively, as controls. For 20 *T. castaneum* only and 20 *T. castaneum* plus 20 g homogenized wheat, headspace samples were collected at 2–4 h for investigation. For 20 *R. dominica* only and 20 *R. dominica* plus 20 g homogenized wheat, headspace samples were collected at 22–24 h for investigation. For 20 *S. granarius* only and 20 *S. granarius* plus 20 g homogenized wheat, headspace samples were collected at 46–48 h for investigation (Table 1). 

### 4.4. Single-Species Experiments

Single-species experiments were performed for the three pest species using 250 mL Erlenmeyer flasks sealed with a Teflon-faced septum lid. The experimental design consisted of a wheat control without insects, and treatments with either insects only or with wheat plus insects (Table 1). Mixed-age and mixed-gender batches of insects were used in all runs. According to our pre-experiments, the amount of the insects used in this research was the limit that can provide a consistent and stable VOCs release. The test insects were transferred on wet filter paper for half an hour allowing them to crawl, cleaning the insect body, and the insects were then cleaned further by transferring them to filter paper. Thirty minutes were allowed for the insects to settle before the flasks were sealed. There were three replicates for each treatment and control. The wheat grains used in all the other experiments were harvested and stored at 10 °C until used. Samples were then transferred into a 25 °C cabinet overnight. The volatile collection method referred to is described in the above “fibre selection”. Periodic blank flasks were tested as procedural controls to confirm the stability and purity of the system. 

### 4.5. GC and GC-MS Analysis

A Varian 3400CX GC (Varian Instruments, Sunnyvale, CA, USA) equipped with a split–splitless injector, a ZB-WAXplus column (30 m × 0.32 mm i.d. × 0.25 μm film thickness), and a Flame Ionization Detector (FID) was used to analyse the volatile profiles extracted by SPME. The gas chromatograph oven was operated under the following temperature program conditions: 35 °C for 8 min, increasing to 120 °C at a rate of 5 °C/min, and held at 120 °C for 10 min. FID temperature was set at 250 °C. Nitrogen was used as the carrier gas, injected at 250 °C in a constant flow of 1.1 mL min^−1^.

Analysis and identification of analytes was carried out on a Shimazu gas chromatograph–mass spectrometer (GC-MS-QP2010 Plus). A Stabilwax^®^ Restek column (30 m × 0.25 mm i.d. × 0.25 μm film thickness) was used with helium as the carrier gas at a constant speed of 30 cm/s. Oven temperature programme conditions included being held at 35 °C for 8 min, increased from 35 to 120 °C at 5 °C/min where it was held for 10 min, then increased from 120 to 245 °C at 15 °C/min where it was held for 10 min. Total run time was 53.3 min. Electron ionization was at 70 eV. Qualitative analysis of the samples was carried out by scanning the mass range between 40 and 300 amu. Peaks were identified by mass spectra comparisons against the standards of the National Institute of Standards and Technology (NIST) spectral library database and by comparison with the retention times of known authentic standards (Sigma-Aldrich, Pty, Ltd., MD, USA).

### 4.6. Data Analysis

The GC analysis data, including the retention time and the peak area, were collected and integrated into the chromatography software, Agilent Chem Station, and then exported to Microsoft Excel and SPSS 20.0 for statistical analysis. The chromatogram pattern features, including detected peak retention times and peak areas, were analysed and compared to verify the repeatability of replicates from the same treatment. The variance between peak areas was analysed using an ANOVA (single factor) with post hoc Tukey’s test by using SPSS 20.0 to compare volatile emissions between different treatments, including: wheat only, wheat plus *T. castaneum*, wheat plus *R. dominica*, and wheat plus *S. granarius*.

## Figures and Tables

**Figure 1 molecules-27-01963-f001:**
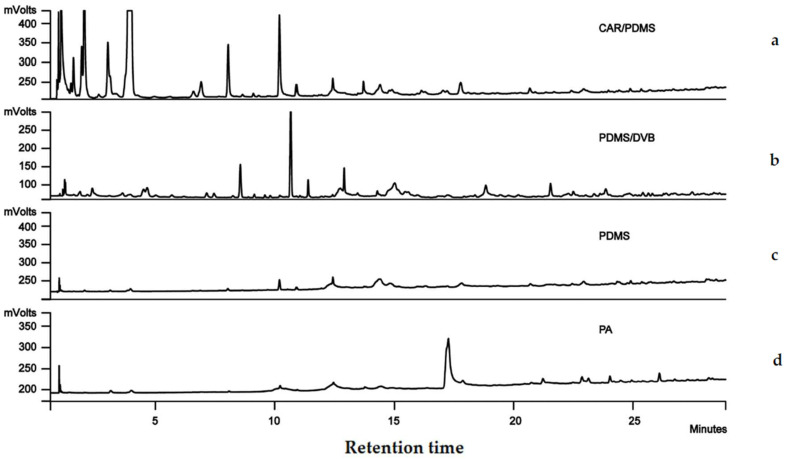
Comparison of different fibres. GC-FID chromatograms of headspace volatiles from wheat extracted using (**a**) CAR/PDMS; (**b**) PDMS/DVB; (**c**) PDMS; and (**d**) PA SPME-type fibres.

**Figure 2 molecules-27-01963-f002:**
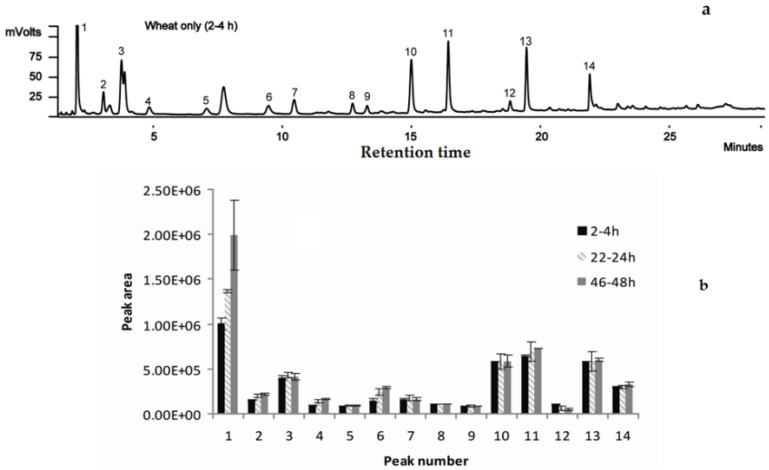
Comparison of mean GC responses of main peaks from healthy wheat collected at 2–4, 22–24, and 46–48 h after being sealed in an airtight container: (**a**) GC chromatograms of headspace volatiles from wheat only and (**b**) the GC peak areas of main peaks from healthy wheat collected at 2–4, 22–24, and 46–48 h after being sealed in an airtight container. Numbered peaks are: 1 = Acetone; 2 = Methanol; 3 = Ethanol; 4 = 2,3-Butanedione; 5 = 2-Butanol; 6 = Hexanal; 7 = 2-Methyl-1-propanol; 8 = 1-Butanol; 9 = 1-Penten-3-ol; 10 = 3-Methyl-1-butanol; 11 = 1-Pentanol; 12 = 3-Methyl-2-buten-1-ol; 13 = 1-Hexanol; and 14 = 1-Pentadecene. Each bar represents the average of three replicates and the error bars indicate standard deviation.

**Figure 3 molecules-27-01963-f003:**
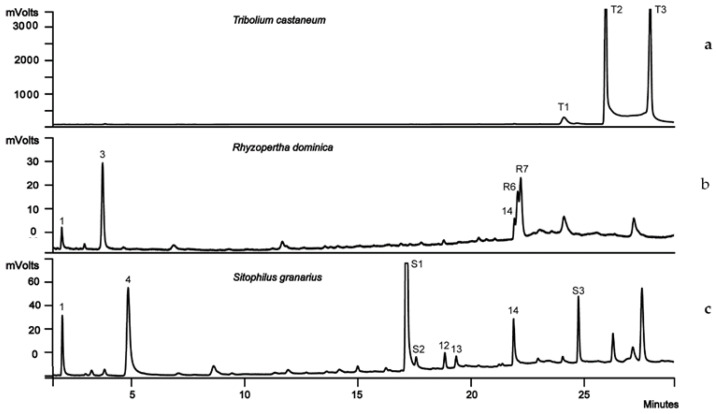
GC chromatograms of headspace volatiles from (**a**) *Tribolium castaneum*, (**b**) *Rhyzopertha dominica*, and (**c**) *Sitophilis granarius*. Peaks T1, T2, T3 are benzoquinones; peak 1 = Acetone; peak 3 = Ethanol; peak 4 = 2,3-Butanedione; peak 12 = 3-Methyl-2-buten-1-ol; peak 13 = 1-Hexanol; peak 14 = 1-Pentadecene; peak R6 = Dominicalure 2; peak R7 = Dominicalure 1; peaks S1–3 = typical peaks of *S. granarius* only; and peak S1 = 3-Hydroxy-2-butanone.

**Table 1 molecules-27-01963-t001:** The seven treatments used in GC, GC/MS analysis.

Sample	Volatile Collection Time (h)
20 g Homogenized wheat only	2–4, 22–24, 46–48
20 *Tribolium castaneum* (Herbst) adults only	2–4
20 *T. castaneum* adults + 20 g homogenized wheat	2–4
80 *Rhyzopertha dominica* (F.) adults only	22–24
80 *R. dominica* adults + 20 g homogenized wheat	22–24
100 *Sitophilus granarius* (L.) only	46–48
100 *S. granarius* adults + 20 g homogenized wheat	46–48

**Table 2 molecules-27-01963-t002:** Volatile organic compounds (VOCs) collected from wheat only, wheat plus *T. castaneum*, wheat *plus R. dominica*, and wheat plus *S. granarius* identified from gas chromatography analysis and Kovats’ values calculation.

Compounds	RT	NIST RI	Kovats	Match Quality	GC Response (10^5^) ± SD, *n* = 4			
	(min)		indices	(%)	Wheat	Wheat + *T. castaneum*	Wheat + *R. dominica*	Wheat + *S. granarius*
**Acetone**	1.18	1116	862	79.5	112.53 ± 6.77	124.36 ± 8.03	17.53 ± 3.35	127.18 ± 9.06
**Methanol**	2.51	1157	901	77.0	32.42 ± 4.29	43.28 ± 5.51	33.09 ± 4.52	32.57 ± 4.19
**Ethanol**	3.48	1215	943	79.1	75.52 ± 6.14	5.049 ± 2.75	49.51 ± 5.02	27.55 ± 3.66
**2,3-Butanedione**	4.43	1248	977	91.8	9.88 ± 2.22	15.49 ± 3.93	65.16 ± 5.15	12.17 ± 3.83
**Ethyl acetate**	5.38	1291	988	85.0	nd	nd	27.16 ± 3.33	nd
**2-Butanol**	7.02	1332	996	71.3	8.83 ± 2.04	7.59 ± 2.09	7.65 ± 2.08	6.94 ± 2.11
**Hexanal**	9.47	1360	1087	77.2	11.27 ± 3.15	19.08 ± 3.31	26.14 ± 3.81	23.51 ± 3.05
**2-Methyl-1-propanol**	10.49	1381	1125	80.1	23.53 ± 3.27	6.86 ± 2.88	7.15 ± 2.11	7.22 ± 2.02
**11-Methylpentacosane**	11.47	1418	1150	89.0	nd	nd	102.27 ± 8.04	nd
**1-Butanol**	12.53	1435	1173	71.9	13.84 ± 4.09	9.55 ± 2.73	9.17 ± 2.50	8.58 ± 2.22
**1-Penten-3-ol**	13.25	1463	1198	97.1	11.62 ± 3.15	8.11 ± 3.11	9.59 ± 2.72	8.06 ± 3.06
**3-Methyl-1-butanol**	14.82	1488	1216	94.3	76.77 ± 5.26	58.92 ± 5.38	61.49 ± 4.85	51.64 ± 5.17
**1-Pentanol**	16.34	1502	1238	84.4	103.52 ± 6.33	95.13 ± 7.09	117.25 ± 8.88	99.64 ± 7.61
**Palmitic acid**	17.57	1527	1251	96.0	nd	nd	55.72 ± 4.92	nd
**3-Methyl-2-buten-1-ol**	18.72	1544	1280	91.0	10.08 ± 2.01	4.73 ± 2.10	22.57 ± 3.27	4.08 ± 2.06
**1-Hexanol**	19.44	1581	1305	75.9	85.83 ± 5.09	78.59 ± 7.77	101.55 ± 8.21	91.21 ± 5.06
**Unknown**	20.81	-	-	-	nd	nd	31.09 ± 4.48	nd
**1-Pentadecene**	21.91	1640	1349	92.2	nd	nd	>250	nd
**Dominicalure 1**	23.58	1685	1377	88.7	nd	nd	26.05 ± 4.17	nd
**Apparent homologs of dominicalure 1**	24.23	1751	1391	94.0	nd	nd	143.95 ± 8.49	nd
**Benzene, 1-ethoxy-4-isothiocyanato-**	24.68	1830	1424	95.7	nd	74.93 ± 6.28	nd	nd
**Stearic acid**	26.11	1892	1451	75.9	nd	68.37 ± 5.92	nd	nd
**Unknown**	27.84	-	-	-	nd	11.5.08 ± 2.50	nd	nd

RT = retention time. NIST RI = retention indices obtained from National Institute of Standards and Technology database (NIST). SD = standard deviation (*n* = 4). nd = not detected.

## Data Availability

The datasets during and/or analysed during the current study are available from the corresponding author on reasonable request.

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
