# Peer review of "Identification of Biomarker Volatile Organic Compounds Released by Three Stored-Grain Insect Pests in Wheat"

_molecules, 2022, doi:10.3390/molecules27061963_

Round 1

Reviewer 1 Report

This is a good study, and the manuscript is well written. I have not more suggestion except the following comments:

L46. Current techniques not only include near infrared spectroscopy, digital imaging and aroma sensing (techniques like electronic nose or “eNose”.

Table 2. Delete internal lines.

L290 I did not get “The insects could clean themselves on wet filter paper for half an hour first and then on dry filter paper for 15 minutes”.

Line 306 delete “.”.

Author Response

Thank you for your response and editing. We have addressed all the comments from the two reviewers. We found the comments helpful and hope that we have incorporated the suggested improvements to your satisfaction. The list of addressed review comments is attached as below in RED fonts.

Reviewer 1

This is a good study, and the manuscript is well written. I have not more suggestion except the following comments:

  1. Current techniques not only include near infrared spectroscopy, digital imaging and aroma sensing (techniques like electronic nose or “eNose”.

Reworded the sentence.

  1. Table 2. Delete internal lines.

Deleted!

  1. L290 I did not get “The insects could clean themselves on wet filter paper for half an hour first and then on dry filter paper for 15 minutes”.

Modified sentence “The test insects were transferred on wet filter paper for half an hour allowing them to crawl for cleaning insect body and the insects were then cleaned further by transferring them to dry filter paper for 15 minutes.”

  1. Line 306 delete “.”.

Deleted

Reviewer 2 Report

Point-by-Point

Molecules – Manuscript # 1635177

Dear Editor and Author(s),

The paper has promising outcomes that will draw readers to the journal. The study uses the HS-SPME technique coupled with GC-MS to investigate volatiles in infested stored grains for biomarkers of insect infestation. Overall, the work is well-written and easy to read. The findings are particularly remarkable by providing a promising source for the diagnosis and monitoring of stored product pests.

I have noticed a few points that need minor adjustments by authors before publication, which are stated below.

Specific remarks.

The introduction highlights the main protagonists in the text and exhibits the potential importance of research. The argument is robust and easy to understand.

Line 73. Change “amongst” to “among”

The Results are remarkably interesting and easy to understand.

Line 102-103 – the authors say that “did not change significantly”. However, there was no mention of a statistical test. Please include or reword statistical information.

Figure 1. letter “c” should be bold.

Line 108-109 – change “(a),” “(b),” “(c),” and “(d),” to “(a)” “(b)” “(c)” and “(d)”. To put it another way, get rid of the comma.

Line 113- change “(a),” “(b),” to “(a)” “(b)”. To put it another way, get rid of the comma.

Line 123 – change (Figure 3) to (Figure 3a)

Line 128 – change (Figure 3) to (Figure 3b)

Line 135 – Include (Figure 3c)

Line 139- change “(a),” “(b),” to “(a)” “(b)”. To put it another way, get rid of the comma.

In the topic Discussion, the importance of results was well-discussed. However, it is important to emphasize the limitations regarding non-laboratory conditions.

Line 236-237 – modify to “However, further work needs to be conducted to determine the detection limits of this technique for R. dominica and S. granaries, mainly in non-laboratory conditions.”

The Material and methods are detailed and easy to understand. However, there is no mention of statistical methods, as pointed out above. Please make it clear. Besides, it could be interesting to incorporate the dataset of pre-test in supplementary methods.

I hope that the comments improve your manuscript,

All the best,

Reviewer.

Author Response

Reviewer 2

The paper has promising outcomes that will draw readers to the journal. The study uses the HS-SPME technique coupled with GC-MS to investigate volatiles in infested stored grains for biomarkers of insect infestation. Overall, the work is well-written and easy to read. The findings are particularly remarkable by providing a promising source for the diagnosis and monitoring of stored product pests.

I have noticed a few points that need minor adjustments by authors before publication, which are stated below.

Specific remarks.

The introduction highlights the main protagonists in the text and exhibits the potential importance of research. The argument is robust and easy to understand.

  1. Line 73. Change “amongst” to “among”

Done!

  1. Line 102-103 – the authors say that “did not change significantly”. However, there was no mention of a statistical test. Please include or reword statistical information.

Revised! “Peak areas of the main compounds detected are very close among the three time points”

  1. Figure 1. letter “c” should be bold.

Done!

  1. Line 108-109 – change “(a),” “(b),” “(c),” and “(d),” to “(a)” “(b)” “(c)” and “(d)”. To put it another way, get rid of the comma.

Done!

  1. Line 113- change “(a),” “(b),” to “(a)” “(b)”. To put it another way, get rid of the comma.

Done!

  1. Line 123 – change (Figure 3) to (Figure 3a), Line 128 – change (Figure 3) to (Figure 3b), Line 135 – Include (Figure 3c)

Done!

  1. Line 139- change “(a),” “(b),” to “(a)” “(b)”. To put it another way, get rid of the comma.

Done!

  1. In the topic Discussion, the importance of results was well-discussed. However, it is important to emphasize the limitations regarding non-laboratory conditions. Line 236-237 – modify to “However, further work needs to be conducted to determine the detection limits of this technique for  dominicaand S. granaries, mainly in non-laboratory conditions.”

Done!

  1. The Material and methodsare detailed and easy to understand. However, there is no mention of statistical methods, as pointed out above. Please make it clear. Besides, it could be interesting to incorporate the dataset of pre-test in supplementary methods.

   Added section 4.6 of “Data analysis” 

Reviewer 3 Report

Comments 

- The authors can find details in the attached pdf file. The authors can also find the copyediting marks at the end of pdf file. When the authors revise their manuscript, the authors must send detailed response for the each of referee comments on the manuscript. When the authors revise their manuscript, the authors must highlight the changes you make in the manuscript by using colored text. 

- The authors need to be consistent with spelling throughout the manuscript and grammar should be checked.

- The authors must supply an ORCID ID for all authors. Getting an ORCID iD is FREE, quick and easy to do through the ORCID registration page: https://orcid.org/register

- The authors must use Journal Guide when they are preparing the revised manuscript.

Author Response

Reviewer 3

- The authors can find details in the attached pdf file. The authors can also find the copyediting marks at the end of pdf file. When the authors revise their manuscript, the authors must send detailed response for the each of referee comments on the manuscript. When the authors revise their manuscript, the authors must highlight the changes you make in the manuscript by using colored text. 

  1. The authors need to be consistent with spelling throughout the manuscript and grammar should be checked.

Done.

  1. The authors must supply an ORCID ID for all authors. Getting an ORCID iD is FREE, quick and easy to do through the ORCID registration page: https://orcid.org/register

Added!